# The Real-Life Impact of mFOLFIRI-Based Chemotherapies on Elderly Patients—Should We Let It or Leave It?

**DOI:** 10.3390/cancers15215146

**Published:** 2023-10-26

**Authors:** Balázs Pécsi, László Csaba Mangel

**Affiliations:** Institute of Oncotherapy, Medical School and Clinical Center, University of Pécs, 7624 Pécs, Hungary; mangellaszlo@gmail.hu

**Keywords:** metastatic colorectal carcinoma, mFOLFIRI-based treatments, cancer in the elderly

## Abstract

**Simple Summary:**

The oncologic treatment of elderly patients is going on with a lack of evidence due to their underrepresentation in clinical trials. Many data suggest that certain groups of elderly patients may benefit from the systemic treatment of their metastatic colorectal tumors. We performed retrospective data analysis to investigate the clinical course of care and clinical outcomes of 515 patients who received first-line mFOLFIRI-based chemotherapy, focusing on a comparison of patients over and under 70 years of age. Considering the PFS and the OS values, only a non-significant trend was observed in OS favouring the younger population. Our conclusion is that patients over 70 years of age with good performance status and controlled co-morbidities benefit from systemic therapy, its modifications and local treatment to the same extent as younger patients.

**Abstract:**

Aim: The oncologic treatment of elderly patients is going on with a lack of evidence due to their underrepresentation in clinical trials. Many data suggest that certain groups of elderly patients, like their younger counterparts, may benefit from the systemic treatment of their metastatic colorectal tumors (mCRC). Method: We performed retrospective data analysis to investigate the clinical course of care and clinical outcomes of 515 patients who received first-line mFOLFIRI-based chemotherapy for mCRC between 1 January 2013 and 31 December 2018 at the Institute of Oncotherapy of the University of Pécs, focusing on a comparison of patients over and under 70 years of age, defined as the cut-off value. Results: 28.7% of the 515 patients were 70 years old and older (median age 73.5 years). Compared to the data of the elderly patients, the younger group (median age 61.1 years) had a performance status that was significantly better (average ECOG 1.07 vs. 0.83, *p* < 0.0001), and significantly more patients received molecularly targeted agents (MTA) (21.6% vs. 51.8%, *p* < 0.0001); nevertheless, mPFS (241 vs. 285 days, *p* = 0.3960) and mOS (610 vs. 698 days, *p* = 0.6305) results did not differ significantly. Considering the 1y PFS OR and the 2ys OS OR values (0.94 [95%CI 0.63–1.41] and 0.72 [95%CI 0.47–1.09], respectively), only a non-significant trend was observed in OS favouring the younger population. Additional analysis of our data proved that the survival in patients over 70 years was positively affected by the addition of MTAs to the doublet chemotherapies, and the reasonable modifications/reductions in dose intensity and the addition of local interventions had similar positive effects as observed in the younger patients’ group. Conclusions: Age stratification of mCRC patients is not professionally justified. Patients over 70 years of age with good performance status and controlled co-morbidities benefit from systemic therapy, its modifications and local treatment to the same extent as younger patients. With the increasing incidence of age-related cancers due to the rising average lifespan, prospective randomised clinical trials are needed to determine the real value of systemic therapy in the elderly and the rational, objective methods of patient selection.

## 1. Introduction

The modern world presents numerous challenges for the constantly evolving healthcare sector. With life expectancy increasing (currently 73.2 years worldwide) and the ageing social structure of developed countries, the proportion of elderly people is also rising rapidly. In developed countries, the proportion of people aged 65 and over is between 20–25%. While the welcome rise in average age is explained by the economic strengthening of society and improvements in healthcare performance, the increase in the proportion of the elderly population is also accompanied by the accumulation of multiple and diverse health and social problems [1].

Colorectal carcinoma (CRC) is the 3rd most common and 2nd deadliest malignancy, accounting for nearly 1.9 million new cases and 0.9 million lives lost each year. The incidence of CRC is increasing by 0.5–1.0% per year, with a current global average of 19.5/100.000. The incidence of CRC in people under 50 years of age is increasing by 2.2% every five years, while interestingly, in people over 65 years of age, it decreases by 3.3%. However, some countries with an exceptionally high human development index could manage to halt the increase in incidence [2,3]. CRC mortality is 9/100.000, i.e., nearly half of the patients finally die of this disease. There has been a slow increase, albeit less than the incidence, mainly due to innovative and more efficient healthcare methods. This negative trend has been reversed in some countries that have introduced different screening methods, with a measurable reduction in mortality of 8–16% [4,5]. The maximum mortality is observed in the Central-Eastern European countries, where the mortality is twice the world average, 20.3/100.000. As the incidence of CRC increases with age, 54.4% of new cases are over 65 years (29.7% over 75 years), and this healthcare challenge is pronounced in developed countries with higher average age [6,7,8]. At the time of detection, 21% of cases are already metastatic, with a higher rate in younger age groups, up to 26%. The rate of synchronous metastatic cases is increasing (5% increase over the last 20 years). In primarily early-stage cases, the chance of developing metachronous metastatic disease is close to 15–20%. Conclusively, more than 40–45% of patients affected by CRC progress to stage IV [9].

Several difficulties can be found in caring for the growing number of elderly mCRC. Firstly, there is a lack of evidence to make appropriate professional decisions, as elderly patients are generally underrepresented in clinical trials. Consequently, the data obtained cannot be fully applied to the care of elderly patients. There is also a lack of complex assessment systems to identify elderly patients with the same tolerance to oncological treatment as younger patients, those who could still benefit from some reduced form of treatments and those whose “frailty” makes active oncological care inappropriate.

For this reason, the collection of real-world data has high priority. We analysed our activities in 6 years of first-line care of mCRC with mFOLFIRI-based treatments. In our previous work, we proved that the definitive dose reductions of the treatments did not negatively impact the outcome data of the patients [10]. This work focuses on the outcomes of care of elderly patients, the impact of different treatment modalities and their modifications on clinical outcomes, compared to the data in younger patients.

## 2. Method

We surveyed all ongoing first-line mFOLFIRI-based treatments at the Institute of Oncotherapy of the University of Pécs Clinical Centre between 1 January 2013 and 31 December 2018. We analysed 25 different patient-related, disease-specific and outcome parameters in every patient (e.g., patient-related data: age at mCRC diagnosis, sex, performance status; tumor-related data: localisation, TN stage, grade of the primary tumor; metastasis-related data: temporality (simultaneous or metachronous), focality and localisation of metastases; treatment-related data: primary tumor resection, type of chemotherapy, number of chemotherapy cycles, metastases ablations; modifications of chemotherapy: drug holiday, dose reduction (rate, timing, length), dose intensity, average cycle-time; clinical outcome parameters: time-on-treatment (ToT), progression-free survival (PFS) and overall survival (OS), the reason for discontinuation of the chemotherapy, the result of the first evaluation of the systemic treatment).

In the current work, based on these data, using 70 years of age as the cut-off limit, the results of 70-year-old and older patients’ group (70<) were compared to the group of patients under 70 years of age (<70). We surveyed the effect of treatment decisions on the different age groups to visualise their real effectivity in the elderly. The study’s primary objectives were the ORR, mPFS, mOS, rate of 1y PFS and rate of 2ys OS.

Descriptive statistics were used to characterise the patient cohorts. Differences in categorical parameters were analysed using a two-sample *t*-test. The level of significance of *p* ≤ 0.05 was used. Progression-free and overall survival were estimated using the Kaplan–Meier method. The odds ratio was calculated within 95% confidence intervals.

It should be noted that at the time of data analysis (17 March 2022), 40 patients (7.8%) were still alive.

## 3. Results

Between 1 January 2013 and 31 December 2018, 515 patients received first-line mFOLFIRI-based treatment for mCRC at the Institute of Oncotherapy of Clinical Centre of the University of Pécs. In addition to comparing the clinical outcomes of the two age groups, 70 years and above (70<) and under 70 years (<70), we investigated the impact of treatment alternatives for elderly patients on clinical outcomes as well. The general information about our patients and the data on their metastatic tumors are collected in Table 1.

In the 70< and <70 groups, no significant difference was found in terms of mPFS (7.9 vs. 9.4 months, *p* = 0.3960) and mOS (20.1 vs. 22.9 months, *p* = 0.6305), with a discrete trend showing, unsurprisingly, an advantage for the younger age group. A comparison of progression-free survival and survival values for age groups can be seen on the Kaplan–Meier curves in Figure 1.

Comparing patients in the 70< group if the cut-off value is 75 years (<75 = 103 patients, or 69.6%, and 75< = 44 patients, or 30.4%), the mPFS were 273 and 264 days (*p* = 0.9686) and the mOS were 673 and 577 days (*p* = 0.8881).

The first evaluation’s RR showed no significant difference. The CR—PR—SD—PD ratio was 0.7–21.9–63.5–13.9% in the 70< group and 0.9–29.7–55.6–13.8% in the <70 group. The RR was 22.6 and 30.6% in each of the two groups (*p* = 0.2121). PD was observed practically at the same rate.

The 1 L treatment was stopped in the two age groups due to physician-guided (objective, e.g., haematological, gastrointestinal) intolerable toxicity in 22.9% and 19.3% (*p* = 0.3562) and due to the patient’s decision (subjective toxicity) in 15.5% and 11.2% (*p* = 0.2016). Any toxicity that impeded the continuation of the therapy appeared in 38.5% and 30.5% (*p* = 0.0806). The difference is insignificant, though the trend shows the moderately enhanced vulnerability of the elderly group.

Analysing the effect of different patient- and tumor-related factors, interestingly, the 70< group had a significantly higher proportion of right-sided tumours (31.1 vs. 19.3%, *p* = 0.0073) and metachronous metastases (37.8 vs. 27.8%, *p* = 0.0253). However, no significant difference was observed in the proportion of single- and multi-organ metastases (74.3/25.7% vs. 70.8/29.2%, *p* = 0.4279) or in the localisation of metastases (liver metastasis in 66.2/64.3%, lung metastasis in 23.6/26.9% and peritoneal metastasis in 16.9/17.7%).

### 3.1. Comparisons Regarding the Type and Dose Modifications of Systemic Treatments

#### 3.1.1. Use of Molecularly Targeted Therapies

MTA complements the standard doublet chemotherapy (mFOLFIRI, FOLFOX6). Significantly fewer 70< patients received MTA supplementation to the doublet regimen (VEGFi (vascular endothelial growth factor inhibitor), 21.6 vs. 51.8% (*p* < 0.0001), and EGFRi (epidermal growth factor receptor inhibitors), 25 vs. 34.2% (*p* = 0.3070). The clinical benefits ratios obtained from MTAs are shown in Table 2.

In both age groups, the difference in clinical outcomes between patients who received MTA (VEGFi or EGFRi) and those who did not did not reach a significant level (PFS *p* = 0.2367 and *p* = 0.1256, OS *p* = 0.2489 and *p* = 0.1067, respectively), although the trend clearly showed a benefit of MTA use. Although there was a robust, improving trend in rates for 1-year progression-free survival and overall survival beyond two years with MTAs, it reached significance only in the <70 group regarding 1-year progression-free survival (*p* = 0.0074). In both age groups, the improvement in overall survival beyond two years with MTA was close to significant (*p* = 0.0669 and *p* = 0.0632). The rates and trends were nearly identical in the two age groups, indicating an age-independent benefit of MTA use.

Nevertheless, we cannot exclude a kind of over-selection for MTA use in the case of elderly patients.

#### 3.1.2. Cycle Number

The median number of treatment cycles was the same in both age groups (12). Still, there was a significant difference in the mean value (13.9 vs. 16.3, *p* = 0.0423), indicating that the younger age group substantially improved long-term treatment adherence, as expected. See Table 3.

#### 3.1.3. Dose Intensity

The planned dose delivery is hampered by the occasional or permanent postponement of treatments (→cycle-time increase) and the occasional or permanent varying rate of dose reduction (→dose reduction). Relative dose intensity (RDI) is the rate between the planned and delivered doses for the same period.

##### Cycle-Time

Cycle time increased from 14 days per protocol to a median of 17.31 days in the 70< group and a median of 16.92 days in the <70 group (*p* = 0.1546) due to various postponements (toxicity, diagnostics, holidays and personal requests). The difference is insignificant and may probably be explained by the more easily manageable toxicity in the younger group. See Table 3.

##### Dose Reduction

Dose reduction occurred in 22 patients (14.9%) in the 70< group and 30 patients (8.2%) in the <70 group (*p* = 0.0416), which is a significant difference. Dose reductions predominantly affected irinotecan dosing, comprising 86.4% in the 70< group (57.9% 10–30% dose reduction and 42.1% complete dose discontinuation) and 93.3% in the <70 group (20–25% dose reduction in 25.0% of patients and 75.0% complete dose discontinuation). The most common cause of dose reduction was persistent neutropenia, sometimes with difficult-to-control nausea/vomiting. There were only 3 vs. 2 cases of EGFRi treatment discontinuation due to intolerable skin toxicity in the two age groups, respectively. See Table 3.

##### Relative Dose Intensity (RDI)

There was no significant difference in mRDI between the two patient groups (*p* = 0.4572). The optimal RDI above 85% recommended in the literature was achieved by 41.2% in the 70< group, compared to 46.6% in the <70 group (*p* = 0.2679). Interestingly, in both age groups, the clinical outcomes of the RDI < 85% subgroup were better than those of the RDI > 85% subgroup. There was a significant improvement in the PFS values; the OS difference in the <70 group was near significant, while in the 70< group only a slight trend of improvement was seen. Comparing the values of the age groups, there was a nearly significant difference only in the RDI > 85% cohort. These results parallel the observations published in our previous work, proving the superiority of lower RDI in PFS/OS in mCRC 1 L treatment in the same patient cohort [10]. See Table 3 and Table 4.

The 1L mFOLFIRI-based treatment was discontinued in the 70< and <70 groups for the following reasons: PD (46.6% vs. 54.8%, *p* = 0.0942), objective toxicity (22.9% vs. 19.3%, *p* = 0.3562), treatment discontinuation by the request of the patient (15.5% vs. 11.2%, *p* = 0.2016), focal treatment of oligometastasis (metastasis ablation) (8.1% vs. 7.1%, *p* = 0.6883), drug holiday (4.1% vs. 6.0%, *p* = 0.3436) and CR (2.7% vs. 1.6%, *p* = 0.4751). Differences in these contexts did not reach the significance level. When objective and subjective toxicity-based discontinuation were assessed together (38.4% vs. 30.5%), more treatment discontinuations occurred in the 70< group approaching significance (*p* = 0.0806), showing elevated fragility in the elderly.

### 3.2. Comparison Regarding Local Forms of Care

#### 3.2.1. Primary Tumour Resection (PTR)

PTR among the synchronous metastatic patients occurred in 75.7% of the 70< group and 71.9% of the <70 group (*p* = 0.3878). Surprisingly, PTR occurred in a minimal but higher proportion of the more sensitive patient group. PTR had a beneficial effect on mPFS/mOS in both age groups; moreover, in the younger population, these advantages proved significant considering all examined survival parameters. Meanwhile, only in mOS was there a substantial difference in the effect of PTR in the elderly age group. The rate of PFS over one year in the 70< group was almost the same as in the group without PTR (33 vs. 36.1%, *p* = 0.7364), and there was no significant difference in the rate of OS over 2-years (43.3 vs. 31.0%, *p* = 0.2412), despite a trend. See Table 5.

#### 3.2.2. Focal Care of Metastases (Metastasis Ablation—MA)

Focal treatment of metastases (surgical metastasectomy, stereotactic body radiotherapy (SBRT), transarterial chemoembolisation (TACE) and radiofrequency ablation (RFA)) was performed in 22.3% of the 70< group and 17.4% of the <70 group (*p* = 0.2679). In the 70< group, there was no significant difference in mPFS, 1-year PFS and mOS between the metastasis-ablated and non-ablated groups (*p* = 0.8923, *p* = 0.7244 and *p* = 0.1255, respectively). Still, the difference in the 2-year overall survival rate was already significant (*p* = 0.0032). In the <70 group, metastasis ablation strongly affected all parameters tested, particularly survival outcomes. Therefore, in a direct comparison of the two age groups, a significant difference was demonstrated between the ablated groups regarding mPFS and mOS (*p* = 0.0485 and *p* = 0.0497) in favour of the younger age group. See Table 6.

### 3.3. Subgroup Comparisons within the 70< Age Cohort

If we create a theoretical reference group within the 70< group, whose members received the minimum treatment, only mFOLFIRI without the addition of MTA, drug holiday, PTR and metastasis ablation, then the value of each treatment modality can be compared to this reference group. In this context, all therapeutic interventions complementing the doublet chemotherapy showed significant benefits in both PFS and OS values. PFS was mainly affected by PTR, meanwhile showing vital significance for OS improvement with all modalities above. The situation of RDI is specific in this context. In contrast to previously suggested dose maximisation in the medical literature, patients with RDI < 85% showed significant benefits in PFS and OS compared to patients treated with RDI > 85%, without valuable dose reduction. Table 7 summarises the clinical value of each therapeutic modality/modification compared to the control group.

In conclusion, despite the poorer PS, higher right-sided tumour rate and significantly lower MTA use in the older age group, clinical outcomes were comparable to the younger age group in our mFOLFIRI-treated patient cohort. Although not equally beneficial, the individual therapeutic modifications driven by the best clinical practice showed similar trends considering outcome data in different age groups. No data suggesting a disadvantage of the treatment of elderly patients were found. See Figure 2.

## 4. Discussion

When an elderly patient is diagnosed with a malignant disease, the question rightly arises: will the patient die of the tumour or with the tumour, or will the treatment of the tumour be fatal? Will the patient tolerate antineoplastic therapy that is thought to be effective? Can it be hoped that the treatment will have more advantages than disadvantages? Can we protect the patient from the expected toxicities of well-intentioned treatment? [11]. These questions must be answered before the patient can be involved in the treatment process.

The care of mCRC in the elderly is fraught with risks. The challenges start with the fact that the concept of ’elderly’ is not clear, and age alone is not representative of the actual condition and health capacity. Due to the strict, sometimes age-related inclusion criteria in clinical trials, the proportion of elderly patients, both socially and disease-specific, differs substantially from the general age structure, with elderly patients being underrepresented. Consequently, the data obtained from clinical studies cannot be fully applied to the care of elderly patients. The application of scoring systems to assess biological, mental, psychological and social capacity to cope, which would allow a near-objective determination of suitability for treatment, is not an obligate part of routine clinical practice. The broadening spectra of innovative systemic oncotherapies and local treatments increase the complexity of the care of elderly patients. The challenges of CRC care in the elderly population are further aggravated by the fact that adjuvant systemic therapies are under-planned and often withheld, considering advanced ages. Hence, the chances of metachronous metastasis are significantly higher.

### 4.1. What Is Considered Old Age?

It is probably impossible to define the concept of old age and an absolute cut-off value. In RCTs and real-world meta-analyses, 65, 70, 80 and even 85 years cut-off values are used. Aging is a non-uniform process that can vary considerably from person to person. Co-morbidities, physical abilities and mental and functional status may vary independently within the same age group. Therefore, chronological age alone is entirely inadequate to determine the actual exercise capacity of a patient. It is probably easier to define the so-called “super-aged” group, as the prevalence of “clinical frailty” or “frail elderly” increases sharply after the age of 85, with interindividual differences decreasing significantly in this age group. The functional reserve capacity of the body is reduced in this age cohort, leading to the development of functional limitations, resulting in a reduced ability to tolerate stress and thus cope with the stress of chronic disease and long-lasting oncological treatment. In the state of the so-called frail elderly, this reserve is critically depleted, with energy sufficient for minimal life support but not for further exertion. Such patients cannot be nominated for stressful, risky, symptomatic treatments, and care may be limited to maintaining organ function [11]. This group should be separated for special oncologic care. Due to the rising average age and advances in health care, most authors agree that age 70 is a realistic cut-off value [12].

#### 4.1.1. The Evidence Is Lacking for the Care of mCRC in Older People

There is a lack of knowledge on treating elderly patients, who are underrepresented in registry studies. The elderly patients included in clinical trials have excellent PS and organ function, and therefore, these patients were not representative of their age group counterparts who are treated in daily practice [13]. The median age in the RCTs that predominantly defined the treatment of mCRC was 60–62 years, whereas the median age of mCRC patients in national databases was 71–74 years. In clinical trials, the proportions of patients aged 65, 70 and 75 years and older were 36%, 20% and 9%, respectively, compared to 60%, 46% and 31% in the US cancer population [14,15,16,17,18], which is a statistically significant (*p* = 0.001) underrepresentation of the elderly age group in trials [19]. Only retrospective clinical trials presented the same age proportion as in real life [20]. The population-specific factors for this underrepresentation were co-morbid conditions, lack of understanding, lack of/inadequate social and home support and unfavourable/uncertain advantage-disadvantage perceptions [21]. Nevertheless, some studies have demonstrated similar survival benefits in older patients without significantly increasing toxicity, although dose adjustments were more often required [22]. However, most of the data on the care of elderly patients comes from analyses of non-interventional or retrospective studies and meta-analyses.

#### 4.1.2. Undertreatment of Elderly Patients

Uncertainty, lack of knowledge and unjustified fear make it challenging to treat elderly patients. In the absence of evidence, choosing a kind of passivity is more accessible than taking the risk of an uncertain outcome. Nevertheless, there has been continuous progress in the care of this age group. From the late 1980s to the mid-2000s, the proportion of palliative oncology treatments increased from 20% to 60% for patients under 75 years of age and 2% to 40% for those over 75 [23]. In the adjuvant setting, the difference is higher, with 79% of patients under 75 years of age receiving postoperative chemotherapy compared to 19% of patients over 75 years of age. A study dealing with the effect of innovative therapies found that patients over 75 years were less likely to receive doublet chemotherapy and MTA therapy (40% vs. 86%). In this study, older patients had a significantly shorter mOS (10.5 vs. 24.5 months) and a higher frequency of toxicity-related hospitalisations (21 vs. 11%). However, patients over 75 years who received combination therapy had an mOS of 21.3 months. The differences between the two age groups were mainly due to underestimating survival probabilities and the lack of effective treatments in the elderly cohort. The authors concluded that the consequence of adjuvant undertreatment in older age is a higher rate of metachronous metastases among patients [24].

#### 4.1.3. Assessment of the Patient’s Actual Capacity

With advancing age, health problems, co-morbid conditions, limited physical abilities, declining cognitive abilities and socio-economic limitations become apparent. Co-morbid conditions can make it challenging to identify new pathologies, which are increasingly atypical in their presentation. As a consequence of these changes, the assessment of the actual capacity to cope is becoming increasingly complex. Still, without a correct assessment, a correct decision cannot be made to start a long-term, burdensome treatment. In addition, actual health capacity changes can be expected during the treatment process, making a re-assessment of capacity necessary [25].

A large meta-analysis of many cases and studies found that treatment outcomes for “frail” patients are less favourable than for less fraught patients. This group had a higher mortality rate, more frequent and severe complications, more postoperative complications and an increased need for transfusion and hospitalisation [26]. Risk factors associated with patient treatment: increased toxicity of chemotherapeutic agents used; failure to receive GCSF/EPO prophylaxis; age over 72 years; aggressive primary tumour type, advanced stage, more organs involved; fragile laboratory values; haemoglobin < 10–11 g/dL, creatinine clearance < 34 mL/min; hearing loss; limited walking distance; need for assistance with basic activities of daily living and medication; reduced social activities and depression. Another study classified patients into three risk groups based on the above, with a strongly significant difference in toxicity between groups [27].

For a rational assessment of capacity to cope, the following factors should be assessed: PS; co-morbidities and their severity, their impact on lifestyle and planned treatment; medications taken, appropriateness of medication, assessment of drug interactions; nutritional status, eating habits; mental and emotional status; socio-economic status; daily activities, self-care; available help; housing and education. A range of assessment methods are available for all of these factors [12].

Patients with impaired organ function are at increased risk of toxicity. Although fundamental pharmacokinetic changes are not observed in an elderly but healthy body, co-morbidities and pathological conditions with advancing age can significantly affect the distribution of drugs in the body (circulatory insufficiency, hypalbuminaemia), activation/degradation (exhaustion of cytochrome P450 system), elimination (reduced glomerular filtration, chronic liver failure) and organ effects of active metabolites (myelosuppression, mucositis, cardiotoxicity, neurotoxicity). Conclusively, it is hard to determine the optimal and safe treatment of co-morbid elderly people with impaired organ function, with dose reduction being justified much more often than in younger ages [28].

#### 4.1.4. The Problem of Choice between Expanding Care Options

Systemic treatments for CRC are in a constant state of flux, with traditional 5-FU-based bolus regimens evolving first into a continuous infusion (CI) and then into doublets and even triplets with the advent of IRI and OXA. The appearance of MTAs has further increased efficacy. The median overall survival of 10.5 months achieved with classical bolus 5FU has now been improved to 30–32 months (TRIBE) with MTA-supplemented triple chemotherapy (BEV-FOLFOXIRI).

The surgical possibilities and technologies for CRC have also improved significantly (e.g., total mesorectal excision, using of staplers, laparoscopy, self-expanding stents), with faster, more thorough, less stressful operations and less perioperative risk. Moreover, surgical treatment of metastases has become routine in high-profile centres. Other local treatment modalities (SBRT, TACE, RFA, MWA) have become available, as well as alternatives or supplementation to surgical treatment, significantly impacting patient outcomes.

### 4.2. Experience with Systemic Treatment of mCRC in the Elderly

#### 4.2.1. 5-Fluorouracil (5-FU)

A meta-analysis based on the results of the largest available 22 RCTs on mCRC treatment in the elderly found that the sole 5-FU treatment, without moderate toxicity differences, yields equivalent mPFS (5.5 vs. 5.3 months, *p* = 0.01) and mOS (10.8 vs. 11.3 months, *p* = 0.31), independently of age cohorts. The only significant difference was observed between 5-FU CI and bolus administration [29]. Another study noticed a significantly higher incidence of diarrhoea, nausea and vomiting in elderly, primarily female patients [30].

#### 4.2.2. Irinotecan

Adding IRI to 5-FU treatment had an equally positive effect on mPFS (under 70 years *p* = 0.0001 and over 70 years *p* = 0.0026) and mOS (under 70 years *p* = 0.003 and over 70 years *p* = 0.15). An expected increase in toxicity was also observed, which did not differ significantly between different age groups, with the following Grade 3/4 toxicity incidence: neutropenia 28.9 vs. 29.7%, diarrhoea 20.5 vs. 23.4%, nausea 11.3 vs. 10.8% and vomitus 9.6 vs. 9.7%, respectively [31].

#### 4.2.3. MTA

A meta-analysis of several RCTs investigating the efficacy of MTAs found that their use in elderly patients had similar excess efficacy compared to younger patients (PFS *p* = 0.017 and OS *p* < 0.0001) [32]. Another meta-analysis measured a considerable difference in mOS (25.8 vs. 22.9 months, *p* = 0.0008) with the same mPFS (10.2 months) values. In patients over 75 years, although there was a clear benefit of using BEV, mPFS and mOS were significantly shorter than in younger patients [33,34]. Several meta-analyses of BEV use in the elderly group (over 65 years of age) have observed the same adverse event frequency when comparing groups receiving and not receiving BEV. An identical arterial thrombotic event frequency (16.4% vs. 17.1%) was observed in the different treatment groups. The stroke frequency was higher in the BEV group (4.9% vs. 2.5%), as expected, but interestingly, the rate of cardiac events was higher in the group without BEV (14.5% vs. 11.5%) [35,36]. Generally, the frequency of adverse events does not increase with BEV administration (135/100.000 vs. 141/100.000 patient days) [37]. In one study analysing the use of CET in the elderly, even though significantly more preexisting co-morbid conditions were observed in the patient group over 65 years, the rate of Grade 3/4 toxicities was 20% in both groups. The incidence of any skin symptoms was 64.2%, also independent of age (*p* = 0.34). No significant difference in PFS was observed (6.5 vs. 7.0 months) [38].

#### 4.2.4. The Treatment Number

A meta-analysis of numerous RCTs suggests that the number of treatment cycles in older age is non-significantly lower, 6.2 vs. 8.3 (*p* = 0.142), despite no significant difference in toxicity. There was a difference in mPFS proportional to the difference in the number of treatment cycles (9.3 vs. 12.8 months, *p* = 0.09), which was also non-significant, while mOS values were nearly the same (24.7 vs. 25.0 months, *p* = 0.208) [39]. In another study, these data were almost identical (mean 7.5 cycles/patient) [40]. A study with a high proportion of patients over 65 years of age, examining the effect of RDI on clinical outcome, found that the median number of treatment cycles in the low and high RDI groups was 7 and 12, respectively [41].

#### 4.2.5. Experience with the Treatment of Senior Patients (over 85 Years)

Several small RCTs and meta-analyses have examined systemic treatment options for geriatric patients. One meta-analysis found doublet treatment feasible in 62.6% of patients over 80 years of age, with a 21-month mOS outcome [42]. Even over 85 years of age, with appropriate patient selection and tolerable toxicity, treatment can be provided with a different but still meaningful efficacy (TTP 8.0 months, OS 15.3 months) compared to younger patients [43]. PS and nutritional status were worse in this group, and more frequent dose reductions were required. At this age, the expected OS benefit was smaller than in younger patients (18.5 vs. 28.8 months, *p* = 0.052). However, the OS of patients receiving chemotherapy was not significantly longer, probably due to the low number of cytostatic components, but it was longer in numerical trend (18.5 vs. 8.4 months, *p* = 0.33) [44,45].

## 5. Conclusions

The care of elderly patients (70<) is hampered by the lack of clinical evidence and the difficulty of measuring actual exercise capacity. Replenishing these data, prospective studies and complex assessment systems are essential for the maximally effective care of the increasing number of elderly mCRC patients.

Our recent observations on the care of elderly mCRC patients proved that the 1 L chemotherapy’s additions and modifications in patients 70< cause changes in clinical outcome with the same trend and magnitude as in patients <70. The difference in tolerance between the two age groups indicates that optimal patient selection is essential due to the increased risk associated with elderly age.

Our study has concluded that complex oncological treatment cannot be decided based on age data alone. However, considering all the elements of complex care, regardless of age, must be beneficial to patients by carefully assessing their capacity to tolerate complex cancer treatments.

## Figures and Tables

**Figure 1 cancers-15-05146-f001:**
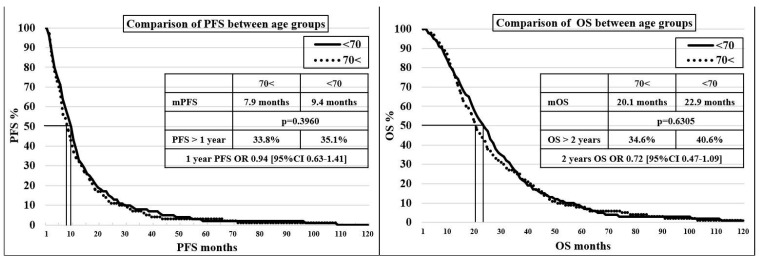
Kaplan–Meier curve for comparison of progression-free survival and survival between age groups.

**Figure 2 cancers-15-05146-f002:**
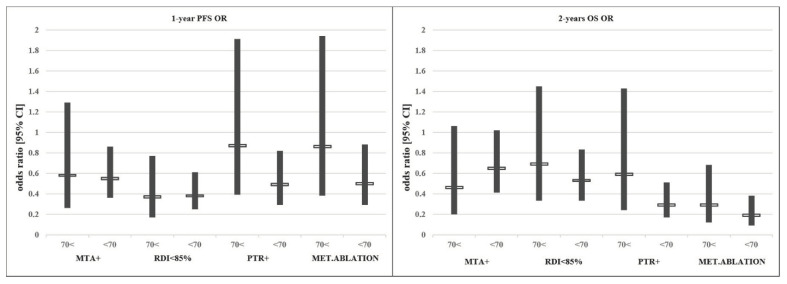
Effect of each treatment modification/addition on the odds ratio (OR) of PFS/OS.

**Table 1 cancers-15-05146-t001:** Patients’ general information.

	70<	<70	*p*-Value
No. of patients	148 (28.7%)	367 (71.3%)	
sex ratio (female/male)	39.9%/60.1%	42.0%/58.0%	NS
median age	73.5 év (70.0–84.6)	61.1 év (26.2–69.9)	
PS (average)	1.07	0.83	*p* < 0.0001
right/left-sided tumor	31.1%/68.9%	19.3%/80.7%	*p* = 0.0073
meta-/synchronous metastases	37.8%/62.2%	27.8%/82.2%	*p* = 0.0253
single/multiple organ metastases	74.3%/25.7%	70.8%/29.2%	NS
rate of liver metastases	66.2%	64.3%	NS
rate of lung metastases	23.6%	26.9%	NS
rate of peritoneal metastases	16.9%	17.7%	NS

**Table 2 cancers-15-05146-t002:** Results of the use of MTAs by age groups.

	MTA+	MTA−	MTA+	MTA−
use of MTAs	21.6%	78.4%	51.8%	48.2%
mPFS days	291	220	304	259
*p*-value	0.2367		0.1256	
PFS > 1 year	43.8%	31.0%	41.6%	28.2%
1y PFS OR	0.58 [95%CI 0.26–1.29]		0.55 [95%CI 0.36–0.86]	
mOS days	766	579	752	617
*p*-value	0.2498		0.1067	
OS > 2 years	55.2%	36.1%	53.7%	43.1%
2y OS OR	0.46 [95%CI 0.20–1.06]		0.65 [95%CI 0.41–1.02]	

**Table 3 cancers-15-05146-t003:** Characteristics of mFOLFIRI-based treatments by age groups.

		70<	<70	*p*-Value
No. of cycles	median	12	12	
	average	14	16	0.0423
median of average cycle-time		17.31 days	16.92 days	0.1546
rate of dose reduction		14.9%	8.2%	0.0416
mRDI		81.48%	83.76%	0.4572
RDI above/below 85%		41.2%/58.8%	46.6%/53.4%	0.2679

**Table 4 cancers-15-05146-t004:** Impact of RDI on the two age groups.

	70<		<70	
	RDI > 85%	RDI < 85%	RDI > 85%	RDI < 85%
	41.2%	58.8%	46.6%	53.4%
mPFS days	176	295	215	333
*p*-value	0.0007		0.0075	
PFS > 1 year	21.3%	42.5%	23.9%	44.9%
*p*-value	0.0070		<0.0001	
1y PFS OR	0.37 [95%CI 0.17–0.77]		0.38 [95%CI 0.25–0.61]	
mOS days	578	659	576	791
*p*-value	0.5872		0.0755	
OS > 2 years	35.3%	44.0%	40.3%	56.2%
*p*-value	0.3324		0.0045	
2y OS OR	0.69 [95%CI 0.33–1.45]		0.53 [95%CI 0.33–0.83]	

**Table 5 cancers-15-05146-t005:** Effect of primary tumor resection (PTR) on the two age groups.

	70<		<70	
	PTR+	PTR−	PTR+	PTR−
	75.7%	24.3%	71.9%	28.1%
mPFS days	244	204	301	257
*p*-value	0.2110		<0.0001	
PFS > 1 year	33.0%	36.1%	39.4%	24.3%
*p*-value	0.7364		0.0063	
1y PFS OR	0.87 [95%CI 0.39–1.91]		0.49 [95%CI 0.29–0.82]	
mOS days	617	578	791	524
*p*-value	0.0146		<0.0001	
OS > 2 years	43.3%	31.0%	56.5%	27.7%
*p*-value	0.2412		<0.0001	
2y OS OR	0.59 [95%CI 0.24–1.43]		0.29 [95%CI 0.17–0.51]	

**Table 6 cancers-15-05146-t006:** Effect of metastasis ablation (MA) on the two age groups.

	70<		<70	
	MA+	MA−	MA+	MA−
	22.3%	77.7%	17.4%	82.6%
mPFS days	287	233	315	266
*p*-value	0.8923		0.0321	
PFS > 1 year	36.4%	33.0%	48.4%	32.3%
*p*-value	0.7244		0.0102	
1y PFS OR	0.86 [95%CI 0.38–1.94]		0.50 [95%CI 0.29–0.88]	
mOS days	832	547	1093	615
*p*-value	0.1255		<0.0001	
OS > 2 years	63.3%	33.3%	79.6%	42.2%
*p*-value	0.0032		<0.0001	
2y OS OR	0.29 [95%CI 0.12–0.68]		0.19 [95%CI 0.09–0.38]	

**Table 7 cancers-15-05146-t007:** Clinical value of each therapeutic modality/modification compared to the control group (MTA−, RDI < 85%, PTR−, MA−) in the 70< age group.

	Control	MTA+	RDI < 85%	PTR+	MA+
No. of cases	29 (19.6%)	32 (21.6%)	69 (46.6%)	112 (75.7%)	33 (22.3%)
mPFS	173	291	308	244	287
*p*-value	1	0.0438	0.0005	0.0064	0.0398
mOS	480	766	725	617	832
*p*-value	1	0.0055	<0.0001	<0.0001	<0.0001
PFS > 1 year	27.6%	43.8%	44.9%	33.0%	36.4%
1y PFS OR		0.49 [0.17–1.43]	0.47 [0.18–1.20]	0.77 [0.31–1.91]	0.66 [0.23–1.96]
OS > 2 years	18.2%	55.2%	49.2%	43.3%	63.3%
2y OS OR		0.18 [0.05–0.67]	0.23 [0.07–0.76]	0.29 [0.09–0.92]	0.13 [0.03–0.48]

## Data Availability

The data presented in this study are available on request from the corresponding author. The data are not publicly available due to personal data protection.

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
