# Peer review of "The Real-Life Impact of mFOLFIRI-Based Chemotherapies on Elderly Patients—Should We Let It or Leave It?"

_cancers, 2023, doi:10.3390/cancers15215146_

Round 1

Reviewer 1 Report

B. Pécsi et al performed retrospective data analysis to investigate the clinical course of care and clinical outcomes of 515 patients who received first-line mFOLFIRI-based chemotherapy for mCRC focusing on a comparison of patients over and under 70 years of age. They found that age stratification of mCRC patients is not professionally justified. Patients over 70 years of age with good performance status and controlled co-morbidities benefit from systemic therapy, its modifications and local treatment to the same extent as younger patients. There were several major limitations in this manuscript.

1. Although the article presents innovative ideas, the manuscript lacks readability due to poor English writing skills. The language used is convoluted and difficult to understand.

2. The table format needs improvement, as it does not follow the standard three-line table format, and the table header is not positioned above the table.

3. Essential patient information is missing from the manuscript.

4. The discussion section does not sufficiently analyze the results presented in the article.

5. The conclusion section is excessively long and fails to highlight the key findings of the article.

6. The reference list requires proper formatting, as some references have inconsistent font sizes.

The English writing is badly. 

Author Response

Dear Reviewer,

We thank You for Your opinion, advices and support.

Here are our reactions to Your comments.

  1. Although the article presents innovative ideas, the manuscript lacks readability due to poor English writing skills. The language used is convoluted and difficult to understand.

Yes, You are right; we have few English writing skills; we are not native English. We really do appreciate to warning us about it. We wrote this article in our language and then made a translation that we considered understandable. Later, AI-supported software helped to make it better, and finally, a native English lector (not a medical professional) finished the whole process. We asked him to review it again. 

  1. The table format needs improvement, as it does not follow the standard three-line table format, and the table header is not positioned above the table.

Thank You very much for this observation. Basically, we used the format of tables that we used before. We didn’t find any requirements about table format in Cancers „Instruction for Authors” section. But we must admit that the format You suggested is more proper and widely used in our literature. All tables were transformed to the required format.

  1. Essential patient information is missing from the manuscript.

During the process of preparing this paper, we had to deal with a large amount of data which was surely too much for the frame of this paper. Seems like we do not always choose well which data to delete. Thank You for warning us about this deficiency in our paper. Thank God, these data exist in a table format, that we, according to Your advice, replace it to its original place in text. I hope that these are the data that You feel missing.

  1. The discussion section does not sufficiently analyze the results presented in the article.

This paper is only a small part of all the data and analysis we did about the problem of 1L treatment of elderly mCRC patients. We had to focus on the most important results that support our basic opinion that elderly patients may be treated the same way as younger patients. We tried to avoid being too detailed, which might disturb the understandability and distract the focus from the main message. We tried to consider carefully all the aspects of treating elderly people; those You find in the “discussion” section are the most relevant factors. We are really grateful for Your opinion, and hopefully, Your advice will help us to achieve a proper paper.

  1. The conclusion section is excessively long and fails to highlight the key findings of the article.

Thank You very much for this remark, Reviewer 2. made the same. We have to admit, that after reading Your advice, we also considered the „conclusion” section too long. Probably, we wanted to explain everything that was in our head and became too long. So we made a significant reduction, keeping the most important message we have and deleting the redundant or obvious information.

  1. The reference list requires proper formatting, as some references have inconsistent font sizes.

Thank You very much for drawing our attention on proper formatting. We reviewed the whole list and made a new re-formatting temporary all fonts seem to be consistent.

Reviewer 2 Report

Dear Authors,

 This is a retrospective study about effectiveness of FOLFIRI as a 1st line chemotherapy for elderly patients with metastatic colorectal cancer. I agree with the opinion that selected elderly patients benefit from doublet chemotherapy with or without molecularly targeted agents (MTA). The manuscript shows a lot of your data. However, I think it contains too much to lead to the conclusion that “FOLFIRI is effective even for elderly patients”. For example, authors showed that “drug holidays” and low relative dose intensity (RDI) positively affected PFS and OS regardless of the age of the patients. Though this is quite intriguing data, it is not related to the theme that “FOLFIRI is effective even for elderly patients”. Therefore, I think that these data and description made the manuscript redundant and exhausting to read through. I recommend the authors to omit irrelevant parts from this manuscript and to write another paper that focus on “drug holidays” and low relative dose intensity (RDI).

Would you concentrate on the theme and make it easier to read.

I have some other comments as follows:

  In page 8 “3.4. Interpreting the results of this study” is redundant.

  In page 9 line 268, “mean” and “average” is usually same meaning. I think “mean” should be “median”, isn’t it?

In page 10 line 334 and in page13 line 503, “ageing” should be “aging”.

  In page 13 to 14, “conclusion” part is too long. I think more succinct description is preferable.

   Regards,

Author Response

Dear Reviewer,

We thank You for Your opinion, advices and support.

Here are our reactions to Your comments.

For example, authors showed that “drug holidays” and low relative dose intensity (RDI) positively affected PFS and OS regardless of the age of the patients. Though this is quite intriguing data, it is not related to the theme that “FOLFIRI is effective even for elderly patients”. Therefore, I think that these data and description made the manuscript redundant and exhausting to read through. I recommend the authors to omit irrelevant parts from this manuscript and to write another paper that focus on “drug holidays” and low relative dose intensity (RDI).

Thank You very much for the time You spent making our paper better. We even had more other points of view for this comparison that we rejected while writing this paper. We have to admit that this is a very complex paper with many information. We focused on the fact that 1L treatment affects the same way in the elderly, and we wanted to show that even the changes of the treatment (dose reduction, cycle-time extension, etc.) act the same way. That is why we consider that RDI belongs to the essential part of our ideas. (We are temporarily working on another paper that only focuses on the effect of RDI in 1L treatment; the results we have are quite surprising.) But the analysis of the effect of “drug holiday” really doesn’t belong to the main line of the paper, so we deleted it from all parts of it. Hopefully, thanks for Your advice the paper will be more understandable and more transparent.

In page 8 “3.4. Interpreting the results of this study” is redundant.

Yes, it is really redundant. You probably know that after reading the text 100 times, things look different than with a fresh and clean head. Thank You for this observation. Longer is not always more, like here and now. 90% of this part was deleted to avoid more redundancy.

In page 9 line 268, “mean” and “average” is usually same meaning. I think “mean” should be “median”, isn’t it?

Thank You very much for pointing out this potential misunderstanding part of the text. Sometimes, the authors’ blindness impedes them from seeing the best solution. Yes, we understand the problem. It was not correctly written, which made it problematic to understand properly. For every patient, we calculated the average cycle-time (from a statistical point of view, it represents more effectively the real difference in cycle-time, than the mean value), and in the sequence of each patient’s average cycle-times in each patient-subgroup we checked the median value. So, in the proper way, I will write „… in the median of average cycle-times …” And after the correction of this misunderstandable part we deleted the whole section according to Your other advice about the redundancy of section 3.4.

In page 10 line 334 and in page13 line 503, “ageing” should be “aging”.

Thank You for Your remark. The mistyping was corrected according to Your advice.

In page 13 to 14, “conclusion” part is too long. I think more succinct description is preferable.

Thank You for this advice also. We have to admit, that after reading Your advice, we also considered the „conclusion” section too long. Probably, we wanted to explain everything that was in our head and became too long. So we made a significant reduction, keeping the most important message we have and deleting the redundant or obvious information

Reviewer 3 Report

The authors reported an interesting topic regarding the real-life impact of mFOLFIRI-based chemotherapies on el-2 derly patients – Should we let it or leave it? The manuscript is well-written and merits publication and meets the readership of cancers for medical and surgical oncologists. However, I would like to suggest the citation of some related articles from the Asian fellow researchers with some discussion to facilitate the comprehensiveness of this important report.

1. Asian Journal of Surgery. Volume 44, Issue 5, May 2021, Pages 715-722

Author Response

Dear Reviewer,

We thank You for Your opinion, advices and support.

Here is our reactions to Your comment.

However, I would like to suggest the citation of some related articles from the Asian fellow researchers with some discussion to facilitate the comprehensiveness of this important report.

Thank You very much for Your support. We really do appreciate the advice You gave. Seeing our „product” from a fellow researcher's eye always makes the quality and scientific importance higher. Reading the paper You suggested came the idea, that only retrospective trials able to produce the same age proportion as the one that can be seen in real life. Another idea You gave is to look for the effect on PFS and OS in different age groups based on a RAS status. In our further research we will focus on this subject as well. Among the 45 citations 19 comes from the last decade, and 7 of them originate from Asian fellow researchers, showing the growing importance of Asian colleagues. On the subject of mCRC, we may not miss their opinion and experiences, their citations belong to the major points of our paper. Thank You.

Round 2

Reviewer 1 Report

The revised manuscript has been impoved greatly, I think it could be published at this version. 

Not applicable

Reviewer 2 Report

Dear Authors,

Congratulations for the amended manuscript. It became succinct and much easier to read. 

Regards,